# Computational Approaches to the Rational Design of Tubulin-Targeting Agents

**DOI:** 10.3390/biom13020285

**Published:** 2023-02-02

**Authors:** Helena Pérez-Peña, Anne-Catherine Abel, Maxim Shevelev, Andrea E. Prota, Stefano Pieraccini, Dragos Horvath

**Affiliations:** 1Department of Chemistry, Università degli Studi di Milano, Via Golgi 19, 20133 Milan, Italy; 2Laboratory of Chemoinformatics, Faculty of Chemistry, University of Strasbourg, 4, Rue Blaise Pascal, 67081 Strasbourg, France; 3Laboratory of Biomolecular Research, Paul Scherrer Institute, Forschungsstrasse 111, 5232 Villigen, Switzerland; 4Department of Biochemistry and Molecular Biology, Universitat de Barcelona, Gran Via de les Corts Catalanes, 585, 08007 Barcelona, Spain

**Keywords:** computer-aided drug design, microtubules, microtubule targeting agents, virtual screening, molecular docking, molecular dynamics simulations, pharmacophore screening, QSAR

## Abstract

Microtubules are highly dynamic polymers of α,β-tubulin dimers which play an essential role in numerous cellular processes such as cell proliferation and intracellular transport, making them an attractive target for cancer and neurodegeneration research. To date, a large number of known tubulin binders were derived from natural products, while only one was developed by rational structure-based drug design. Several of these tubulin binders show promising in vitro profiles while presenting unacceptable off-target effects when tested in patients. Therefore, there is a continuing demand for the discovery of safer and more efficient tubulin-targeting agents. Since tubulin structural data is readily available, the employment of computer-aided design techniques can be a key element to focus on the relevant chemical space and guide the design process. Due to the high diversity and quantity of structural data available, we compiled here a guide to the accessible tubulin-ligand structures. Furthermore, we review different ligand and structure-based methods recently used for the successful selection and design of new tubulin-targeting agents.

## 1. Introduction

Microtubules (MTs) are an essential part of the eukaryotic cytoskeleton and are implicated in various diseases. They are highly dynamic polymers composed of α,β-tubulin dimers in which each monomer is able to bind GTP. GTP hydrolysis is limited to the β-monomer (E-site), providing energy for conformational changes required for MT formation. Within the α-monomer GTP is always retained (N-site). Together, these proteins form hollow, cylindrical structures, in cells mostly containing 13 protofilaments. Within the cell, they are involved in numerous cellular processes such as cell signaling, morphology, motility, growth, and long-distance trafficking regulation [1].

Naturally, any perturbation of the MT network severely affects cell survival, thus making MTs attractive targets for cancer therapy. Presently, several MT targeting agents (MTAs) such as vinca alkaloids and taxanes are used to treat different types of cancer. By altering the MT homeostasis, they promote apoptosis of cancer cells via several independent mechanisms [2]. Moreover, there is an increasing interest in MTs as a target for the treatment of diabetes [3]. Furthermore, abnormal dynamics of MTs in neuronal cells is implicated to play an important role in several neurodegenerative diseases (reviewed in [4]).

Almost 40 years after the first mechanism was proposed [5], the details of MT formation still remain an ongoing topic of discussion; the main steps as understood today are outlined below: Nucleation of MTs occurs in cells at MT organizing centers (MTOCs) such as the γ-TuRC complex (reviewed in [6,7,8]). Based on this template structure, MTs grow by addition of a dimer carrying GTP in both nucleotide binding sites in a head–to-tail fashion, always adding α-tubulin onto exposed β-tubulin. Thus, the MT is formed as a polar structure and exposes β-tubulin at the growing end (MT plus end). Incorporation of tubulin dimers into the MT lattice is accompanied by a conformational change of the dimer from a curved towards a more rigid, straight structure (curved-to-straight transition), which is then followed by GTP hydrolysis in the β-monomer [9]. Only at the plus end of the MT a so-called “GTP-cap” consisting of dimers that contain GTP in both sites is sustained, which is thought to stabilize the end against depolymerization [10].

Within cells, the MT cytoskeleton is maintained in what is termed the “dynamic equilibrium”, alternating between phases of growth and shrinkage of individual MTs, which allows them to perform their various physiological activities (Figure 1). MT associated proteins, post-translational modifications, as well as small molecules MT targeting agents (MTAs), modulate the dynamics of the MT network. MTAs at high concentrations exert different mechanisms of actions, which are used to categorize them into two classes: MT stabilizing agents (MSAs) that lead to an increased stability of the present MT by promoting assembly or stabilization of the lattice structure, and MT destabilizing agents (MDAs) which prevent the assembly of dimers into MTs. 

MTAs have been widely studied and characterized due to their long-standing use as anti-cancer drugs. Routinely, MTAs are probed on their cytotoxicity and their ability to influence MT polymerization. Further, to understand their mode of action a lot of effort has been dedicated to solving high-resolution MT and ligand–tubulin complex structures. Up to 2021, seven distinct binding sites for small molecules had been thoroughly characterized by X-ray crystallography. In 2021, a combination of crystallographic fragment-based screening and molecular dynamics (MD) simulations evidenced 10 binding sites occupied by 56 chemically diverse fragments, of which six sites were completely novel [11]. A selection of these fragments was subsequently used in a straight-forward fashion to develop a lead-like molecule from non-cytotoxic building blocks. It was named todalam and occupies the 8th binding site on tubulin located at the inter-dimer interface [12]. Together, the large amount of biochemical data and ever-growing amount of structural data available lay a solid foundation for the computer-aided development of novel tubulin-targeting agents.

Computer-aided molecular design methods, such as ligand-based and structure-based approaches, open new possibilities to further exploit current knowledge on MTs, tubulin and MTAs. These two in silico strategies have been considered essential for accelerating the research of MTAs assisting in the identification, design, and selection of new compounds. Both are used to discover molecules with desired biological activity, but differ in terms of the initial information exploited to generate their predictions. Ligand-based methods “learn” from previously discovered ligands of a target, and their measured affinities. They are agnostic in terms of ligand-target interaction mechanisms, but rely on interpolation and extrapolation of predicted affinity of a new candidate based on the nearest known examples of ligands. On the contrary, structure-based approaches base their predictions on explicit modeling of presumed interactions between ligands and given biological targets.

The aim of this review is to summarize recent applications of state-of-the-art methods of both computational ligand and structure-based approaches to successful design of new MTAs. Note, however, that using in silico methodology to “discover” putatively active compounds makes no sense unless those compounds are actually synthesized and tested. Publishing in silico predictions without further validation should, in our opinion, be strongly discouraged, because the likelihood of experimentalist readers embarking on the difficult task of synthesis and testing of someone else’s predictions is very low (actually null, as far as we can tell). Therefore, this work will only cite computer-aided design work which is either (a) methodologically innovative, (b) reporting tool benchmarking studies or (c) backed up by experimental validation.

## 2. Ligand-Based Approaches

Ligand-based strategies may be employed if rich and balanced structure-activity information (at least ~100 known tested small molecules, including binders and non-binders to the target) is available. They are of course the only option if no structure of the target protein has been solved, but are irrespectively useful in the early stages of a virtual screening (VS) campaign, as they are typically much faster than structure-based algorithms. These methods algorithmically analyze molecules encoded by molecular descriptors or ensembles of calculated conformations and extract chemical knowledge to predict a given compound’s property. Such screening usually highlights structural patterns deemed important for exhibiting a desired property.

Historically, these methods were the first to be applied to the problem of discovering novel modulators of tubulin polymerization. This was mostly due to the low quality of tubulin-related structural data at that time (reviewed in [13]). However, despite considerable progress in tubulin crystallography and prevalence of structure-based methods in modern tubulin research, ligand-based approaches are still useful and yield promising results. This section highlights recent examples of successful application of such computational methods in tubulin-related drug design.

### 2.1. Similarity Search

A similarity search is used to filter a set of molecules, in search of those that display similar features to a query molecule. This method assumes that similar molecules exhibit—statistically speaking—similar properties [14]. There is no absolute best way to encode molecular similarity, typically rendered by the metric (distance) of the two points representing molecules in “descriptor space”. Fragment-based fingerprints (monitoring the presence of specific substructures in each molecule) are common molecular descriptors for this task; however, other features such as descriptors of molecular shape, topological pharmacophores can be used. Any function that measures distance between two points in a metric space is applicable to characterize “molecular dissimilarity”. The best combination of descriptors and metric function is the one that guarantees the best “Neighborhood Behavior Compliance”, e.g., by minimizing the occurrence of “property cliffs”—pairs of compounds perceived as highly similar in spite of using widely different property values [15].

A similarity search is often used as a first step in VS. For example, Aoyub et al. [16] and Guo et al. [17] performed 2D similarity searches in large compound databases as initial phases of drug design cycles that resulted in development of novel MTAs binding to the taxane and colchicine site, respectively. Several novel colchicine-site targeting agents were also discovered by Mangiatordi et al., who based their design on a 3D shape similarity screening [18]. Another two colchicine-site targeting hits were found by Federico et al., who used not only 3D shape, but also electrostatic potential similarity in their VS campaign [19].

Coupling known active compound structures with information on their targets can make the similarity search useful for establishing targets of novel compounds. This was demonstrated by Lo et al., who developed chemical similarity networks based on two and three-dimensional compound similarity (CSNAP2D and CSNAP3D, respectively). By calculating similarities of molecules with cytotoxic action of unknown mechanism to molecules within the network, the authors correctly predicted tubulin as a target for 37 novel compounds targeting the colchicine and taxane binding sites [20,21].

In Table A1 (Appendix A) we have summarized the implementations of the technique used in mentioned references.

### 2.2. QSAR Modeling

Quantitative structure-activity relationship (QSAR) modeling finds a mathematical function that relates chemical structure to values of some desired property, e.g. biological activity. The process of fitting such a function is called model training. Typically, two- or three-dimensional molecular structures are digitally encoded by various descriptors, which are then input to machine learning algorithms along with corresponding target property values, available from biological assays. These values can be continuous (pIC_50_ values, binding affinity) or discrete (active/inactive classification), corresponding to either regression or classification problems. Afterwards, a trained model can be used to predict target values for new molecules, not included in the training set. The predictive power of a QSAR model depends on careful curation of input data, rigorous validation, and adequate assessment of its applicability domain. State-of-the-art approaches in these topics are described in more detail in [22,23,24].

This method is particularly useful for rational drug design as it provides insight into which molecular features correlate the most with changes of desired property values. For example, Gaikwad et al. used two-dimensional QSAR modeling to establish structural patterns that significantly correlate with cytotoxicity of colchicine site-targeting phenylindoles against cancer cells [25]. High utility of QSAR modeling in VS was demonstrated in works by Guo et al. [26] and Stefanski et al. [27], who used consensus QSAR modeling in VS campaigns that yielded a total of three novel colchicine site targeting tubulin polymerization inhibitors.

3D QSAR was shown to be a convenient way to rationalize ligand optimization in works by Quan et al. [28] and Pandit et al. [29]. Both works used CoMFA and CoMSIA methods to rationalize structure-activity data for limited datasets of similar scaffold-based compounds, suggesting possible structure optimization patterns, which, in case of the latter work, yielded a new class of cytotoxic in vitro tubulysin derivatives targeting the vinca binding site. A summary of the experimental conditions for the above-mentioned QSAR works is provided in Table A2.

It is worth noting that the use of machine learning in this field has been limited due to the scarcity of publicly available data. The lack of large, diverse tubulin-related structure-activity datasets makes it difficult to train adequate machine learning models that can be used in a large-scale virtual screening context. For example, querying the ChEMBL database (v.26) for “Tubulin” returns more than 8000 raw structure-activity records, but these are a heterogeneous collection of results from widely different assays at diverse experimental setups, using the MTs or tubulin of widely different species (from *Arabidopsis* to *Homo Sapiens*). Or, machine learning requires homogeneous, comparable experimental activity entries to serve for calibration of empirical functions trying to approximate them upon input of a molecular structure. Thus, only entries sourcing from a same experimental setup (listed under a same ChEMBL Assay ID) can be safely compared. Deceivingly, there is only one such assay (CHEMBL817769; Inhibition of tubulin polymerization interacting at the colchicine binding site of *Sus Scrofa*) featuring more than 100 entries (103, precisely)—a rule-of-thumb minimal threshold of training set size to start envisaging machine learning. Size is necessary, but far from sufficient—a balanced presence of active and inactive compounds is of paramount importance, whereas the chemical diversity of the compounds sets the limit for the applicability domain of the model. Machine learning is likely to play a more prominent role in this regard if more relevant data becomes publicly available.

### 2.3. Pharmacophore Screening

A pharmacophore is an abstract description of the set of local steric or electronic properties (hydrophobicity, H-bond acceptor/donor features, charged groups) that a molecule should contain in order to interact with a particular biological target at a specific site. A set of such properties, with defined positions in space relative to each other is called a pharmacophore model. For a given ligand, it is mostly related to fragments of chemical structure and is binding site-specific. It is assumed that molecules that follow the same pharmacophore pattern may have similar biological activity (even though they may differ in other, less relevant structural aspects). This makes pharmacophore-based VS useful for searching and designing new drugs, escaping the rather narrow domain accessible by strict similarity-driven searching.

In particular, experimental structure-activity data can be used to automatically construct ligand-based pharmacophore models. A detailed explanation of pharmacophore model generation steps is given by Giordano et al. [30]. In short, models are obtained by computing and aligning 3D conformations of selected molecules, with pharmacophore features assigned to overlapping structural fragments. Several models may be built for different alignments. A fitness function estimates how well the molecules fit into a given model, leading to selection of the best model.

Screening with such models can be used to filter compounds in a large library, leaving only those that match the required model in at least one of several conformations. Models always need to be validated before use in VS. A model is considered valid if it can discriminate known active molecules from decoys—structurally similar compounds not showing the desired activity [31].

Ligand-based pharmacophore screening is often used in combinations with other computational methods to lower the number of candidates that need to be tested by subsequent approaches. For example, Zhang et al. used a pharmacophore model based on taxane-site ligands to reduce the number of compounds processed by structure-based pharmacophore model and protein-ligand docking, eventually leading to a discovery of two novel tubulin-targeting cytotoxic agents targeting this site [32]. In a similar manner, a ligand-based pharmacophore model developed by Lone et al. was shown to be useful for vinca-site targeting agents design [33]. Moreover, Niu et al. successfully applied a ligand-based pharmacophore model to discover two novel colchicine-site targeting modulators of tubulin polymerization [34]. Stefanski et al. used a ligand-based pharmacophore model in a VS campaign that discovered two potent in vitro cytotoxic colchicine-site targeting agents [27].

As can be seen, despite ligand-based pharmacophore screening not being featured in many recent tubulin-related computational studies (structure-based pharmacophores or docking being preferable, as soon as experimental protein structures are available), it is still a viable method that is used to design and screen for novel modulators of tubulin polymerization. Table A3 provides an overview of recent works that used this approach.

## 3. Structure-Based Approaches

Contrarily to ligand-based methods, structure-based approaches exploit the 3D structure of a macromolecular biological target to estimate a given molecule’s affinity to a targeted binding site. The main sources of information for these methods are either experimental data generated by X-ray crystallography, NMR spectroscopy, cryo-electron microscopy or computationally predicted data. Analyzing bound ligand poses helps to determine the key residues defining the binding site, as well as pinpoint to the key fragments of molecular structure that contribute to interaction with the target protein. Success in high-resolution determination of biological macromolecule structures drove the usage of these structure-based techniques in modern drug discovery pipelines, and tubulin-related research is no exception. In this section, we review recent examples of structure-based methods application in search and design for novel modulators of tubulin polymerization [35,36].

### 3.1. Structural Data on Tubulin

#### 3.1.1. Tools to Study Tubulin 3D Structures

Possibly, the most important decision in carrying out a structure-based drug design project on tubulin is the selection of the correct tubulin model. While the sheer abundance of accessible information is a huge benefit for any of such projects, the numbers and diversity of available structures can be overwhelming. In order to select the best possible model for one’s purpose, it is important to consider the method and system in which the structure was obtained. Therefore, we will give a brief overview of the available structures and setups that were used to determine them, as well as highlight a few key points to consider when selecting the structure.

By comparing the different structures obtained of tubulin and MTs, it was observed that tubulin dimers are able to adopt two prominent tubulin conformations that are related to its assembly state: a “straight” conformation is present in assembled MTs and a “curved” conformation is observed in soluble tubulin. The conformational transition from curved-to-straight is needed to establish lateral tubulin contacts between protofilaments in MTs. This curved-to-straight transition requires rearrangements of the tubulin monomers, in which the intermediate domain of the tubulin monomer moves with respect to a larger ensemble comprising both the N- and C-terminal domains. Due to this repositioning within the straight MT lattice, the α monomers are almost perfectly aligned with the β monomers, thus it is possible to superpose α onto β simply by translation (Figure 2A). Whereas, within the soluble dimer there is an intrinsic curvature of one monomer against the other, thus translation alone is not sufficient to superpose one monomer onto another (Figure 2B). The degree of this curvature varies; it can range from 9–18 degrees depending on the binding partners present [37].

This conformational state is one of the main differences observed between all available crystal structures and the CryoEM data on MTs: All crystal structures depict the soluble and “curved” conformation of tubulin and all MT structures show the "straight” conformation. Thus, it is important to consider on which “state” of the tubulin structure is used, as basis for the computational work. Despite these major differences, the crystal structures are remarkably well suited for the design and optimization of drugs. Up to now, five different systems have been described for the crystallization of tubulin. All rely on proteins stabilizing the tubulin in its dimeric or tetrameric form, as the uncoordinated, soluble tubulin is polymerizing rather than forming nicely diffracting crystals. This is highlighted by the fact that the first high-resolution crystal structure has only been reported after the tubulin–stathmin interaction had been discovered and exploited [38,39].

The very first structural information on tubulin was obtained in 1998 by Nogales et al. using electron crystallography on taxol stabilized zinc-induced protofilaments. This allowed the determination of a first model of the structure of tubulins, the assignment of domains and identified the taxol binding site on β-tubulin [40]. However, the arrangement of the protofilaments in this crystal system is antiparallel and does not reflect the protofilament-assembly found in MTs. Accordingly, this system was not further used for X-ray crystallographic studies.

Soon afterwards, the tubulin stathmin-like domain SLD (T_2_R) system was the beginning of tubulin complex crystallization with the first crystal structure in 2000 [41], followed by the first tubulin-small molecule complex in 2004 [42], which revealed the position of the colchicine site. Later, it was noted that cleavage of the C-terminal tubulin tails increases the resolution of the T_2_R system significantly. Furthermore, this system evolved to be the most commonly used T_2_R-tubulin tyrosine ligase setup (T_2_R-TTL, Figure 3A) [43,44], which was used to solve most tubulin-small molecule structures. In both complexes, two tubulin dimers are coordinated by a stathmin-like protein RB3 that prevents tubulin polymerization by its N-terminal β-hairpin cap bound to α1 tubulin. In the T_2_R-TTL system, the TTL protein is bound at the same end of the tetramer on α1 tubulin. The overall tubulin structure does not differ significantly between the two setups.

Since the SLDs and TTL used in these crystallization systems may prevent binding of proteins to tubulin, alternatives have been developed. The tubulin Designed Ankryin Repeat Protein DARPin crystallization system (Figure 3B) [45] is the second most frequently used one. This system allows to achieve even higher resolution compared to the T_2_R-TTL one, with the best resolved structure ranking at 1.5 Å resolution (PDB ID 6S8K, [46]). In this system, only one tubulin dimer is coordinated by the selected DARPin, resulting in a much more densely packed and smaller unit cell.

Up to now, the described systems T_2_R, T_2_R-TTL and TD1 are the only ones that have been used to elucidate the structures of tubulin-small molecule complexes. Nevertheless, the following two crystallization systems for the study of protein-protein interactions have been included to provide a complete overview of tubulin crystallization systems.

In order to investigate the interaction of the cellular MT growth factor, Stu2p, Ayaz et al. co-crystallized its tumor overexpressed gene domain TOG1 with tubulin [47]. Surprisingly, it was found that TOG1 was establishing interactions with both α- and β-tubulin and preferentially bound to the curved state of soluble tubulin dimers (Figure 3C).

More recently, a fifth crystallization system, targeting MT binding proteins, has been introduced. Therein, one artificially designed α-Rep protein is used to prevent tubulin polymerization and to enable crystallization of the complex (Figure 3D). α-Rep was specifically designed to bind to tubulin sites involved in longitudinal protofilament interactions in order to expose the surface of tubulin, which would be on the exterior site of the MT [37]. So far, the system has been used to elucidate the structural details of centrosomal P4.1-associated protein CPAP [48], allowing a more throughout investigation compared to the previously published CPAP –tubulin DARPin structures [49,50].

#### 3.1.2. Binding Sites on Tubulin

As mentioned in the introduction, extensive work has been done on determining the binding mode of tubulin-targeting agents. Here, we would like to give a brief overview of the eight established binding sites (Figure 4) and their mode of action on modulating MT dynamics (in more detail reviewed in [51]). The most prominent member of MTAs is paclitaxel, sold as a blockbuster drug under the name Taxol®, which is an MSA that binds to an exposed pocket on β-tubulin. **Taxane-site** ligands are able to enhance MT stability, either by promoting the curved-to-straight transition, e.g., paclitaxel [52,53] or by direct structural stabilization of the βS7-βH9 loop (M-loop), a key structural element forming inter-dimer contacts in MTs [54], e.g., epothilone A or zampanolide [44]. **Laulimalide-/Peloruside-site** agents strengthen the interactions of tubulin dimers across neighboring protofilaments in MTs by binding to a pocket near the lateral protofilament interface. Moreover, these agents have been described to allosterically stabilize the M loop to some extent [55,56].

In the group of MDAs, **colchicine-site** ligands are present with a great variety and a high number of different scaffolds. They bind in a buried pocket at the intra-dimer interface of α and β tubulin, flipping the βT7 loop out of its native position. By occupying this binding site, they effectively prevent the curved-to-straight transition by blocking the compaction of the pocket formed by the strands βS8 and βS9, and by the helices βH8 and αH7 [42,57].

Another well-known group of MDAs are the vinca alkaloids, which bind at the longitudinal interface between tubulin dimers. **Vinca-site** ligands induce a ‘wedge’ [58] at the tip of the MT and thus prevent the straightening of the dimers. Additionally, they promote the assembly of small helical tubulin polymers, thereby effectively reducing the amount of assembly-competent tubulin. It has also been noted that vinca-site ligands interfere with the hydrolysis of GTP by blocking the proper alignment of the catalytic residues, thereby further hindering the polymerization process [59,60].

The group of **maytansine-site** ligands blocks the assembly of MTs by inhibiting the addition of new tubulin dimers to the growing end. This is achieved by binding to the exposed site of β-tubulin and then effectively blocking the site that should accommodate the αH8 and αT7 loop of the binding tubulin dimer [61]. Ligands bound at this site not only block further growth of MTs, but are also capable of fully blocking the formation of smaller tubulin oligomers, at high concentration, effectively keeping tubulin within the dimeric state.

So far, the only ligand known to exclusively bind to α-tubulin is **pironetin**, which binds to a buried pocket by covalent attachment to Cys316 [62,63]. Binding of pironetin perturbs the above-mentioned helix αH8 and the αT7 loop, thus similar to maytansine preventing the interaction of these elements with the neighboring tubulin and fixing tubulin in an assembly-incompetent state. Furthermore, pironetin also prevents the growth at the −end of the MT, which exposes the α-tubulin surface harboring both the helix αH8 and the αT7 loop and thus eventually promotes the disassembly of already formed MTs [62].

Recently, both the 7th and the 8th distinct binding sites on the tubulin dimer have been described. **Gatorbulin**, a cydodepsipeptide isolated from marine cyanobacteria, was found to bind to the intra-dimer interface adjacent to the well-known colchicine binding site [64]. **Todalam**, the first rationally designed tubulin binder, which emerged from a crystallographic fragment screen [11], binds at the inter-dimer interface at a site located between the maytansine site on β-tubulin and the end of the pironetin pocket on α-tubulin [12]. Both compounds are thought to hinder MT formation by a mechanism similar to that of the vinca-site ligands, by creating a wedge into the tubulin-oligomer structure. As observed for vinblastine, todalam as well was shown to promote the formation of ring-like tubulin oligomers, further decreasing the pool of tubulin available for polymerization.

The position of the binding sites has clear implications on the choice of the crystallization system: due to its size, the TD1 crystallization system is well suited for molecules bound internally within one dimer (e.g., colchicine, gatorbulin), however the binding sites at the inter-dimer interface such as for example the vinca-site can only be targeted by using the T_2_R(-TTL) systems.

#### 3.1.3. System Selection for Virtual Screening (VS) and MD Simulations

Not all out of the more than 300 crystal structures within the PDB database were equally often used in computational experiments, as we noticed in our analysis of the most recent MD simulation literature (overview in Table A6). Surprisingly, we found that even 20 years after the first description of the tubulin structure at near-atomic resolution [54], simulations of taxane-site ligands or apo tubulin are often based on some of the very first tubulin datasets obtained with electron diffraction in 1998 (PDB ID 1TUB, 3.7 Å, [40]) and 2001 (PDB ID 1JFF, 3.5 Å, [54]). There is a bit more of variety in the colchicine site structures that were selected for simulations, although only a fraction of the great number of available high-resolution tubulin colchicine site structures have been considered: PDB ID 1SA0 2004 3.6 Å [42], PDB ID 1Z2B 2005 4.1 Å [58], PDB ID 3E22 2008 3.8 Å [59], PDB ID 3HKC 2009 3.8 Å [57], PDB ID 4O2B 2014 2.3 Å [65], PDB ID 6Y6D 2020 2.2 Å [66]. For simulations of other ligands, since a lower number of structures is available, the choice of the starting model was obvious: vinca-site ligands PDB ID 3E22 2008 3.8 Å [59], PDB ID 4O4J 2014 2.2 Å [56], PDB ID 5JH7 2016 2.2 Å [67], and laulimalide site: PDB ID 4O4H 2014 2.1Å [56].

While this analysis reflects on only a fraction of the most recent literature, we see a trend that not always the most recent or high-resolution structures are selected. Due to the importance of the selection of the starting model for virtual screening and MD simulations we provide in Table 1 an overview of the highest resolution structures available to support the selection process. Further, in Table 2 we have compiled a list of the CryoEM models for MT structures with highest resolution for tubulin-small molecule complexes, a field in which not many structures are available yet.

When choosing the VS system, one should also consider the target of the desired molecule. If one is aiming for an MT-binder, one might compare the binding pocket found in crystallization systems with the CryoEM MT structures to evaluate the differences and the impact of MT formation on the specific binding site. However, one needs to be careful because most of the structures have been obtained by stabilizing the MT with small molecules, most often paclitaxel, or using non-hydrolyzable nucleotides. Therefore, these structures could also be different from the MT structure in the absence of stabilizers or artificial nucleotides.

The next consideration on the selection of the system for MD simulation is the assembly of tubulin into protofilaments and MT structures. If the binding site studied is far from any tubulin inter-dimer interface (e.g., colchicine site, gatorbulin) or is considered to completely prevent the interaction of two dimers (e.g., maytansine site, pironetin site), a dimer can serve as a model for tubulin binders. It can be extracted from either T_2_R, T_2_R-TTL or TD1 structures, however the presence of the stabilizing proteins could artificially modify the tubulin structure in the proximity of their binding site. Ideally, the site of VS should be far from crystal contacts established in the system and the binding sites of the stabilizing proteins DARPin, RB3 and TTL.

If the binding site is present at the longitudinal inter-dimer interface (e.g., vinca, todalam, gatorbulin) or the lateral axes (e.g., taxanes), a more complex system may need to be considered. To extract two dimers in the curved conformation either T_2_R or T_2_R-TTL structures can be used to generate longitudinally linked tetramers. In the case of both longitudinal and lateral axes as present only within the context of an MT, a CryoEM structure should be used as a basis. For example, scientists such as Castro-Álvarez et al. [80] opted to study a ‘tetramer’ model to investigate binders at the taxane site, since the M loop stabilized by some taxane-site ligands is establishing lateral interactions with the neighboring tubulin dimer. The choice of the system size is a trade-off between the accuracy of the site and the computational effort needed.

### 3.2. Tubulin-Related VS Strategies

#### 3.2.1. Pharmacophore Screening

We already discussed ligand-based pharmacophore modeling and its application in VS, where models are generated from structures of active molecules relying on conformational space sampling and ligand alignment. In structure-based pharmacophore modeling, a ligand’s bioactive conformation in the binding site along with knowledge of the receptor structure guides the pharmacophore features placement and often provides higher quality models than those deduced by the ligand-based approach [31].

It is common to start such modeling by choosing one or several protein structures with bound ligands. Then, possible interactions are estimated between ligand and binding site atoms. After that, pharmacophore features are automatically assigned to regions of binding site space based on estimated H-bond formation, charge, and hydrophobic contact. Such models can be combined by merging over common features or refined manually [81]. The same validation strategy is applied before usage in VS, as described for ligand-based models.

Structure-based pharmacophore screening has shown significant value in tubulin-related research. It has been mostly used as one of the steps in multi-step VS campaigns that yielded novel colchicine and taxane-site targeting modulators of tubulin polymerization. Interestingly, recent successful works used different approaches to model building and selection. As such, Nagarajan et al. [82] built six colchicine-site interaction models based on relevant crystal structures and merged them by common features to obtain a model later used in a VS. Mangiatordi et al. [18] built seven colchicine-site models based on manually selected relevant PDB structures, validated them with a set of actives and decoys, and used the model with the best discriminative performance for VS. On the contrary, Zhou et al. [83] built four pharmacophore models based on relevant well-resolved PDB structures containing colchicine-site ligands and refined them manually, putting emphasis on interactions with experimentally known key residues. Similarly, Zhang et al. [32] derived seven pharmacophore models of the taxane site interactions from a single PDB crystal structure and refined all of them to highlight only the most important features. However, Gallego-Yerga et al. [84] noted that defining a single pharmacophore model puts unnecessary constraints on the model. Instead, they used an ensemble of 118 pharmacophore models derived from all resolved structures of tubulin with different bound colchicine-site targeting ligands in an attempt to capture flexibility of the site and variating nature of ligands. By contrast, Elseginy et al. [85] was able to produce good results by using a single model automatically extracted from a relevant colchicine site structure without any additional refinement. Table A4 provides an overview of pharmacophore screening implementations from each mentioned VS campaigns.

#### 3.2.2. Protein-Ligand Docking

One of the most frequently used structure-based drug design methods is protein-ligand docking. It is used to estimate with a considerable degree of accuracy the most likely conformation of a ligand within a given binding site, and therefrom extrapolate—with, unfortunately, not very good accuracy—its binding affinity.

By computationally predicting the binding affinity of tubulin-targeting agents, researchers identify compounds that have a high binding affinity for tubulin and are therefore more likely to be effective binders. This information can be used to prioritize compounds for further experimental validation, such as performing in vitro or in vivo assays to confirm their binding activity and efficacy. It’s worth noting that computational predictions of binding affinity are not always accurate, and experimental validation is needed to confirm the predictions. However, computational predictions can be very useful for rapidly and efficiently identifying potential binders and prioritizing them for further experimental validation. Then, the success rate can vary depending on several factors, such as the quality of the computational method, the quality of the input data, and the complexity of the system being studied.

Protein-ligand docking tools operate on 3D structures of proteins and ligands. Typical docking computations involve sampling of a ligand’s conformational space, and ranking the computed poses by estimating the (free) energy of interaction between the ligand in a given pose and the binding site using specific scoring functions. These computations may consider the binding pocket’s residues to be rigid or flexible. Rigid docking is computationally faster, but unable to account for ligand-specific adjustments of the protein site geometry.

Algorithms for conformation sampling modify torsional, translational, and rotational degrees of freedom of a given ligand in a site in either a systematic sequential or a stochastic randomized fashion. Detailed reviews of sampling methods were compiled previously for example by Sulimov et al. [86] or Halperin et al. [87].

Sampling algorithms visit many putative poses of a ligand within the site and the docking software ranks all of them according to a scoring function. These functions aim to estimate a ligand’s affinity toward the binding site in each specific sampled pose, taking into account intermolecular interactions and other physicochemical effects. The calculations are based on either force fields, modeled contribution of empirically defined physicochemical parameters, or knowledge of different atom-type interactions statistically extracted from resolved co-crystallized protein-ligand structures.

Before use, protein structures are pre-processed by adding missing hydrogens, computing charges, removing solvent molecules, ligands, and other heteroatoms. It is considered good practice to validate the suitability of a chosen docking software to model a desired binding pocket, which is most often done by re-docking. It consists of removing a native ligand from the modeled system and placing it back using the docking method of choice. If the best pose output by the software matches the bioactive pose of the native ligand, it is assumed that both the conformation sampling algorithm and the scoring function adequately describe the modeled system and can be used to model interactions of novel ligands with the pocket [88,89].

With protein-ligand docking being an efficient and quick way to obtain significant intuition for drug design and optimization, it has been used in several contexts of tubulin-related drug design. For example, it is often included in VS campaigns as one of the last steps to prioritize a virtual hit for further investigation. As such, Mangiatordi et al. used protein-ligand docking to further filter the results of a prior pharmacophore screening and prioritize remaining compounds, the latter containing 31 novel colchicine-site targeting agents with in vitro anti-proliferative properties [18]. In a similar manner, Guo et al. reported protein-ligand docking as an essential step that allowed them to discover eight confirmed cytotoxic agents targeting the colchicine binding site [26]. Moreover, Zhou et al. used protein-ligand docking to highlight five virtual hits found by pharmacophore screening as most promising ones, their cytotoxic action related to binding at colchicine site was later confirmed in vitro [83]. A work by Ayoub et al. showed how docking-based optimization of VS hits could benefit from pose rescoring using the MM/PBSA method [16].

A noteworthy work by Zhang et al. compared five docking programs by re-docking 10 complexes of tubulin co-crystallized with taxane-site targeting ligands and selecting the three best software programs for evaluation of virtual hits found by pharmacophore screening; among the prioritized molecules, two were established as cytotoxic agents, supposedly targeting the taxane binding site [32]. Protein-ligand docking was instrumental in highlighting 15 virtual hits found by pharmacophore screening in the work by Nagarajan et al., later experimentally confirmed to be cytotoxic in vitro due to targeting the colchicine site of the tubulin protein [82]. Similarly, Federico et al. used docking to evaluate potential affinity of found virtual hits toward tubulin’s colchicine site, eventually discovering seven micromolar inhibitors of tubulin polymerization [19]. Consensus docking of pharmacophore screening virtual hits helped Elseginy et al. establish four novel compounds with significant antiproliferative activity against cancer cells due to targeting the colchicine site of the tubulin protein [85]. Interestingly, Mao et al. incorporated protein-ligand docking and interaction fingerprint similarity comparison to discover a novel taxane-site targeting promoter of tubulin polymerization [90]. Lastly, Stefanski et al. also combined docking and fingerprint similarity measure of protein-ligand interactions as a last step of a VS campaign that yielded two potent in vitro cytotoxic colchicine-site targeting agents [27].

Protein-ligand docking is a powerful VS tool that alone can produce high-quality results. For example, Zúñiga-Bustos et al. used only protein-ligand docking to screen a large compound library, with virtual hits being confirmed promotors of tubulin polymerization targeting the laulimalide binding site [91]. In another study, Liu et al. screened a large database with consecutive docking experiments with increasing rigor of conformational sampling, eventually yielding six hits with in vitro antitumor activity due to targeting the colchicine binding site [92]. In a similar manner, Liu et al. docked a large compound library and discovered two colchicine-site targeting in vitro inhibitors of tubulin polymerization among the highest ranked molecules [93].

Often, protein-ligand docking is used as a way to provide rationale for a tubulin-targeting agent’s biological action. In such case, designed molecules are docked into one or several potentially targeted binding sites. Best estimated poses are then examined in terms of docking scores and physicochemical interactions within the site. Such analysis may also provide ideas for further compound optimization. For example, docking studies were used to assess possible binding modes and guide rational design of colchicine-site targeting compounds of different classes independently reported by Ameri et al. [94], Guo et al. [17], Riu et al. [95], Patel et al. [96], and Mustafa et al. [97]. In a similar manner, Tripathi et al. [98], Ayoub et al. [99], and Chávez-Estrada et al. [100] used protein-ligand docking to estimate putative binding modes of taxane-site targeting molecules. Interestingly, Forero et al. [101] predicted possible binding modes of the designed compounds by docking them into both colchicine and taxane site, eventually settling on colchicine site as the possible target of the designed compounds based on interaction analysis. Finally, Pandit et al. [29] used docking to evaluate binding regimes of vinca-site targeting peptides. Table A5 provides an overview of exact implementations of docking protocols used in mentioned works.

### 3.3. Molecular Dynamics (MD) Simulations to Study Tubulin-Ligand Complexes

#### 3.3.1. Classical MD Simulations Used on Tubulin

Molecular dynamics (MD) is a computational simulation technique that allows exploration of the behavior of a molecular system over time by solving Newton’s equations of motion. This is of great importance for research, as biomolecules are dynamic entities whose atoms are in constant motion. In this way, by using MD, time-dependent processes in molecular systems can be monitored to facilitate the analysis of their structural, dynamic, and thermodynamic properties. 

MD simulations can provide valuable information that is not accessible from experiments, allowing the formulation of new hypotheses. In addition, technical progress, both in algorithm efficiency and computational power, allows the study of biological macromolecules of larger dimensions on longer timescales, and the predictions that are inferred from these simulations make MD simulations a very valuable computational approach in the drug design field. 

MD is widely used as a computational technique to examine protein-ligand complexes, such as the binding of molecules to tubulin and MTs, to analyze the effects on the tubulin structure upon ligand binding.

In the study of MTAs in complex with tubulin using classical MD simulations, different settings need to be considered during system preparation. For instance, the choice of the force field that best suits the system under study is important, since the quality of the MD simulations results depends on the quality of the energy function used to treat the interactions among atoms in the system. Additionally, the simulation time and the MD engine used are important factors that also condition the accuracy of the simulations.

In this review, Table A6 summarizes the settings used by scientists to set up classical MD simulations to investigate tubulin-ligand complexes. Due to the number of articles related to this topic published since 2019, we have decided to dedicate the review of classical tubulin MD simulations to the articles which were published in the last three years and thus are the most up-to-date manuscripts.

By analyzing Table A6, we can observe that most often the tubulin-ligand complex systems are simulated under periodic boundary conditions, solvated in explicit water (TIP3P or SPC water model) in cubic or octahedral box at room temperature and atmospheric pressure. The typical simulation time is ~100 ns. While different force fields are explored, the most prevalent are Amber Force Fields FF99SB and the more recent one FF14SB.

#### 3.3.2. Enhanced Sampling Methods

Enhanced sampling algorithms have appeared as a powerful tool for increasing the efficiency of classical MD simulations. During a certain simulation time, enhanced sampling methods allow for the sampling of larger areas of a complex system configuration space. The accuracy of the results is highly dependent on the selection of the simulation settings. Here, we outline three different enhanced sampling methods used to study tubulin-ligand binding mechanisms.

## 4. Umbrella Sampling (US)

Umbrella sampling (US) is an enhanced sampling computational technique applied to expand the sampling of a system in which ergodicity is hampered by the form of the energy landscape of the system. US is used to calculate the thermodynamic parameters for the binding of a ligand to a protein. In the tubulin field, US has been used to predict the strength of binding (binding energy) of a ligand to tubulin by slowly pulling away the ligand from the binding site. ΔG_bind_ derives from the potential of mean force (PMF), obtained from a series of US simulations. Several initial positions of the ligand with respect to the protein of interest are generated, each corresponding to a location where the ligand is harmonically restrained at increasing center of mass (COM) distance from other selected groups via an umbrella biasing potential. These restraints allow the ligand to sample the conformational space in a defined area along a single degree of freedom (reaction coordinate) [102].

US is subject to certain limitations, such as biases in sampling due to improper selection of reaction coordinates (RCs), challenges in identifying appropriate RCs for complex systems, the need for multiple RCs in systems with multiple reaction pathways, and the method being dependent on the choice of RC. Additionally, the method can be computationally expensive and limited to systems with multiple reaction pathways and high-dimensional systems.

Zhang et al. used US simulations to retrieve the free energy potential of αβ-tubulin separation upon binding to a certain ligand [103]. Also, Zhou et al. and Mane et al. simulated the αβ-tubulin dissociation free energy under different system conditions using the US method [104,105].

## 5. Steered Molecular Dynamics Simulations (SMD)

Steered molecular dynamics (SMD) is another enhanced sampling method in which an additional external force is applied to one or more atoms in the studied system to maintain the constant speed of motion along a selected coordinate [106]. SMD emulates atomic force microscopy (AFM) experiments. It allows the study of molecular processes, such as the protein-ligand unbinding mechanism, by focusing on selected degrees of freedom. It is important to keep in mind that in SMD the force applied is not necessarily proportional to the binding free energy, as it aims to simulate the process of binding a molecule to another, rather than the equilibrium state of the bound complex.

Rai et al. performed SMD to study the bonding strength between eribulin and tubulin isotypes to which it presented the highest (aVIIIbIII) and lowest (aIbII) binding energies, which were previously calculated computationally. They kept the tubulin structures fixed by setting position restraints on their heavy atoms, whereas the eribulin structure was dynamic. They observed that a three-fold greater force was required to pull out eribulin from the active site of one tubulin isotype in comparison to that of another isotype [107].

## 6. Metadynamics (MetaD)

Metadynamics is an enhanced sampling technique that enables conformational sampling of the free energy landscape of a system through the use of collective variables that describe it. Castro-Álvarez et al. used MetaD to study the effect in the tubulin M loop on the binding of laulimalide and peloruside A to the taxane site [80].

Binding pose metadynamics (BPMD) allows for the assessment of the stability of the ligand in solution. This is because BPMD can differentiate between stable and unstable binding geometries. It is expected that the unstable ligand poses will rarely be occupied in the energy landscape under MetaD simulation bias. As a result, unstable ligand poses make a minimal contribution to binding affinity.

Boichuk et al. applied BPMD to evaluate the stability of a colchicine binder in complex with tubulin and to select its most stable conformation using as collective variables the RMSD values of the heavy atoms of the ligand [108]. Fusani et al. compared the binding mode of epothilone A in complex with tubulin of the first published 3D structure solved by Nettles et al. (PDB: 1TVK) and a later one solved by Prota et al. (PDB: 4I50) using BPMD. Fusani et al. wanted to differentiate between the correct and incorrect ligand binding poses by applying BPMD [109].

Moreover, Gaspari et al. used MetaD to induce the *cis*-to-*trans* isomerization of a colchicine binder in complex with tubulin. This allowed the authors to calculate the difference in binding free energy between the *cis* and *trans* isomers of the ligand via a thermodynamic cycle. Furthermore, Gaspari et al. also used MetaD to gain insight into the differences in the unbinding process of colchicine and another colchicine site binder studied in complex with tubulin [110].

When using MetaD as an enhanced sampling method, it is important to be aware of its limitations, particularly in relation to the selection of the collective variable (CV). These limitations include potential bias in sampling, challenges in identifying appropriate CV for complex systems, increased computational cost for high-dimensional systems, and limitations in exploring the free energy surface.

## 7. Applications of MD for Tubulin-Ligand Studies

### 7.1. Docking Validation and Refinement

MD is often used as a post-processing technique to validate and refine the binding modes of the protein-ligand complexes obtained from docking experiments. MD applied for docking validation has also been used in the tubulin research field.

For example, Hadizadeh et al. investigated the possible binding mode of an active tubulin binder (9IV-c) that showed high activity against human tumor cell lines. For this, they used computational methods such as docking and MD. First, they docked 9IV-c in the colchicine site, and the output was later submitted to MD simulations to evaluate and refine the docking results. The simulation of the complex was analyzed using root mean square deviation (RMSD), radius of gyration (Rg), and hydrogen bond stability values. In this way, they obtained a successful prediction of the way 9IV-c binds to tubulin, allowing them to conduct further computational studies to identify new potent tubulin inhibitors [111].

El-Mernissi et al. designed four new colchicine site binders using 3D-QSAR models and docking based on a series of 2-oxoquinoline arylaminothiazole derivatives that were identified as promising tubulin inhibitors. Among the four newly designed binders, MD simulations of the compound with the best docking score were performed to validate its docking binding pose using the RMSD, root mean square fluctuation (RMSF), Rg, and solvent accessible surface area (SASA) metrics. By performing MD simulations, they confirmed the conformational stability of the complex, thus validating their docking experiments [112].

Zhang et al. performed VS using a combination of molecular docking methods of 50 compounds in the taxane site to search for novel tubulin polymerization inhibitors. Subsequently, the best hits were submitted to IC50 experiments, from which the two compounds with the highest antiproliferative activity were selected for MD simulations along with the tubulin-paclitaxel complex. By performing MD simulations, they further studied the binding mode, stability, and molecular interaction pattern of the docking results. Apart from using RMSD, RMSF, and Rg as MD analysis metrics, they performed clustering analysis to extract information on how tubulin in complex with the three studied taxane-site binders is sampling the conformational space. They used ‘BitClust’ [113], which is a relatively new faster implementation of the Daura et al. clustering algorithm that performs rapid structural clustering of long trajectories [114]. In this way, using MD simulations, they validated the stability of tubulin in complex with the two compounds and probed the mechanism of their interactions, which aligned with the experimental results [115].

Elhemely et al. observed that a meta-substituted 3-arylisoquinolinone that had shown a high cytotoxic effect in several cancer cell lines mimicked the structure of colchicine. They hypothesized that its mode of action could be related to its binding to the colchicine site of tubulin. To test the suitability of the compound to bind to this site, the authors first performed docking experiments, which were later refined by MD. These computational studies suggested that the meta-substituted 3-arylisoquinolinone was able to bind well to the colchicine binding site [116].

### 7.2. Comparison of the Binding Free Energy of Different Ligands

The resulting trajectories from MD simulations are also used to compute the free energy of binding of different molecules binding to the same site to obtain a quantitative measure to compare and rank the best hits normally resulting from docking studies. There are different methods to estimate the free energy of binding of protein-ligand complexes such as Free Energy Perturbation (FEP), Molecular Mechanics Generalized-Born Surface Area (MM-GBSA), and Molecular Mechanics Poisson-Boltzmann Surface Area (MM-PBSA). Due to the numerous computational resources required for the performance of MD simulations, this approach can only be used to rank a low number of molecules, in the tens range.

Elhemely et al., in the article mentioned above, computed the free energy of binding applying the MM-GBSA method using the MD-based refined complexes of two 3-arylisoquinolinones bound to tubulin that only differed in the location of a substituent in their structure (meta versus para). The authors wanted to investigate how the change in the substituent position could alter the free energy of binding and compare the binding mode of the molecules in the tubulin sub-pocket. The computational results aligned with the experimental ones, concluding that the meta-substituted molecule was a better colchicine site binder than the para-substituted compound [116].

Stroylov et al. used FEP calculations based on MD simulations for predicting tubulin-ligand free binding energy differences of new tubulin polymerization inhibitors targeting the colchicine site [117].

Mao et al. with the goal of discovering new tubulin inhibitors capable of binding to the taxane site, performed a VS of ~1.6M molecules retrieved from the ChemDiv database. After applying different computational filters, 17 hit compounds were selected and submitted for experimental evaluation. The in vitro tubulin polymerization assay found P2 to be the most promising compound. Therefore, P2 was submitted to MD simulations not only to further investigate the interactions between P2 and tubulin based on the docking results but also to compare it with paclitaxel, an already known active taxane-site binder. They calculated the free energy of binding of both complexes using the MM-PBSA method obtaining—68.25 ± 12.98 kJ mol^−1^ for the tubulin-P2 complex and—146.05 ± 16.17 kJ mol^−1^ for the tubulin-paclitaxel complex. These results were in line with the experimental evidences, defining P2 as a lead compound that could be used for new tubulin inhibitors drug design campaigns [90].

### 7.3. Identification of Key Binding Site Residues

MD is also used to further investigate the mechanisms of interactions between tubulin and hits, as previously reported, and to find key amino acids in the protein that are especially important for binding to the studied ligand within a given tubulin binding site, also called ‘hot spots’.

Neto et al. studied a series of chalcones predicted to bind to the taxane site using both experimental and computational approaches, including MD simulations. To identify the key binding site residues establishing the strongest interactions with the studied ligands, the authors performed Computational Alanine Scanning (CAS) of each tubulin-ligand interface. This allowed analysis of the free energy contribution of the amino acids located at the taxane site, bringing new insights into this tubulin site for further exploitation using chalcones [118].

Gamya et al. reported a noscapine derivative (VPN) discovered and validated using computational tools such as docking and MD simulations. VPN was able to be properly accommodated in the colchicine site according to the docking results, which were then submitted to MD studies for validation of its stability at the site by calculating the RMSD and RMSF values, and its binding free energy using the MM-GBSA and MM-PBSA methods. Furthermore, they performed a deeper analysis of the interactions established between the residues of the receptor with the ligand by calculating the energy contribution of each residue in the binding of VPN by performing Per Residue Energy Decomposition (PRED) analysis using the MM-GBSA method. In this way, they were able to identify the residues that have the greatest impact on the binding and stability of VPN, the ‘hotspots’ [119]. Other researchers have also applied PRED analysis to the search for ‘hotspots’ to investigate the details of tubulin-ligand interactions at the atomic level [90,120].

### 7.4. Analysis of Local and Global Effects upon Ligand Binding

Structure-based computational approaches have also been used to investigate the effect of different MTAs on the local geometry of tubulin. Moreover, since MTs are formed by allosteric proteins, the effect of binding of a ligand at one site can also cause non-local effects in MTs, and therefore, the study of global effects caused by ligand binding is also important.

For example, the M loop has been widely studied by X-ray crystallography and other structural techniques to understand the effect of taxane site binders on this loop [44,70]. This is due to the fact that the M loop is found at the β1/β2 interface and is involved in the stability of the interaction. However, the dynamics of M loops remains unclear, and other research groups approach these questions using SB computational techniques. Castro-Álvarez et al. performed MetaD simulations of laulimalide and peloruside A to analyze the changes produced in the M loop upon binding of these ligands [80]. MetaD helped explain how laulimalide and peloruside A shift the M loop to an α-helix structure by bringing together different residues at the external site of β1.

Basu et al. studied the collective changes that the tubulin over-stabilizing agents paclitaxel and taxotere induce on the structure and dynamics of the α,β-tubulin dimer by performing MD simulations. To study the conformational effects of tubulin induced by the binding of the ligands, they also performed MD of the apo protein to compare the results of the simulations of apo tubulin with those of holo tubulin. They investigated the influence of ligand binding on the essential dynamics of tubulin using Principal Components Analysis (PCA). They observed that the apo tubulin samples a broader range of conformations than that of the holo tubulin. Therefore, the presence of the ligands biases the system toward a more stabilized conformation. Moreover, for a more local structural exploration, the authors performed a Define Secondary Structure of Proteins (DSSP) analysis to study the conformational changes of the M loop and its associated regions induced by the binding of the two ligands. More computational analyzes were performed to thoroughly investigate the effect of binding of both paclitaxel and taxotere on the dimeric structure, concluding that these ligands enhance the α,β-tubulin dimer to be more favorably accommodated into the MT superstructure [121].

### 7.5. Exploration of Ligand Binding to Different Tubulin Isotypes

The α and β tubulin in eukaryotes consist of isotypes that differ in their aminoacidic sequence. Therefore, in the field of tubulin, researchers study not only the binding of different ligands to the same binding site of a certain tubulin isotype, but also the binding of the same ligand to different tubulin isotypes [122]. In silico approaches have a great advantage in the study of tubulin isotypes, since they are rarely accessible to be investigated experimentally. In silico strategies allow for the analysis of the sensitivity of a certain ligand to bind to tubulin isotypes which would be highly demanding to do experimentally. Rai et al. performed MD simulations of the potent anticancer drug eribulin bound to different tubulin isotypes to report differential binding affinities. However, it remains to be explored how the residue composition at the binding site between tubulin isotypes translates into major changes in the tubulin conformation and the binding affinities with ligands [107].

### 7.6. MD Analysis Metrics

As previously described, MD simulations have multiple applications in the *in silico* study of tubulin-ligand complexes. To extract the information of interest from the output of MD simulations (trajectory), different analysis metrics are available. In Table 3 we present the techniques that have been used in the selected tubulin-related articles to analyze MD simulations of tubulin and its interactions with MTAs.

## 8. Conclusions

In this review, we provide an overall picture of the different ligand and structure-based computational methods that have been used in recent years for the study of tubulin-targeting agents, and an overview on the available MT and tubulin structural data. We observed that computer-aided methods have had significant contribution to the field of tubulin-targeting drug design. VS of compounds, applying both ligand and structure-based approaches, provided many hits with in vitro bioactivity. An advantage of ligand-based methods is their computational efficiency and ability to work with big data. They are often beneficial to the early stages of VS, where the goal is to filter out compounds irrelevant to the task at hand in a fast manner. These initial results are well suited for subsequent filtering by structure-based methods, which provide more intuition behind the physico-chemistry of potential interactions between a given virtual hit and the desired biomolecular target. Computational methods were also shown to guide in rational design and optimization of novel tubulin-targeting agents.

Moreover, despite the large number of available tubulin binding sites (8), our analysis shows that the colchicine and the taxane sites are the most studied ones in tubulin-related computational research while the rest are underrepresented. We also observed a tendency to mainly use structure-based methods to find tubulin-targeting agents such as molecular docking for VS and MD for the refinement of the resulting docking hits.

MD simulations have widely been used in the tubulin-directed drug discovery field. In the recent literature, there is a tendency to use MD as a computational docking post-processing method that allow the validation and refinement of the docking results, the analysis of the ligand–tubulin dynamics and the estimation of binding free energies.

We expect growth of interest in these computationally understudied sites in the near future since computational strategies are becoming essential in the first steps of the drug design campaigns.

## Figures and Tables

**Figure 1 biomolecules-13-00285-f001:**
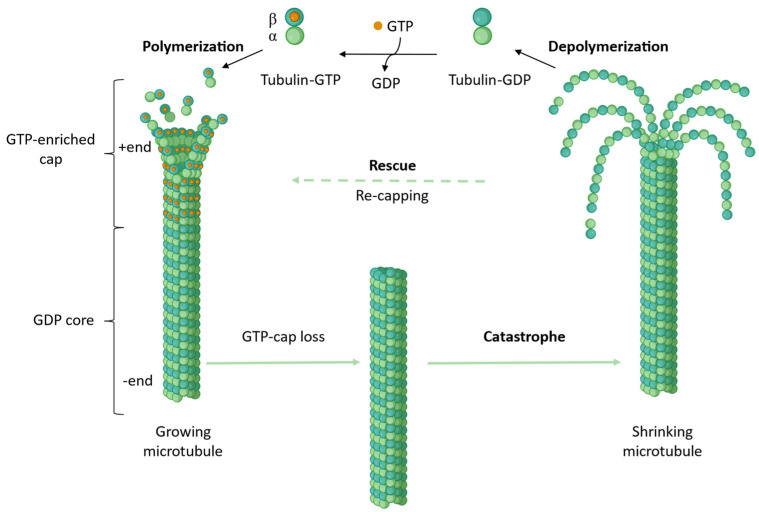
Microtubule dynamic equilibrium. MTs are constantly alternating between growth and shrinkage phases, while the −end of the MT is displaying some dynamics the overall stability is governed by quicker processes at the MT +end. Growth of an MT is facilitated by incorporation of two GTP containing tubulin dimers onto the +tip, followed by lattice incorporation, which leads to subsequent GTP hydrolysis. On the top of the growing MT a “GTP-cap” consisting of GTP-dimers stabilizes the structure. Exchange of this capping dimers against GDP tubulin leads to depolymerization. Adapted from “Microtubule (polymerizing and depolymerizing)” by BioRender.com (accessed on 15 December 2022).

**Figure 2 biomolecules-13-00285-f002:**
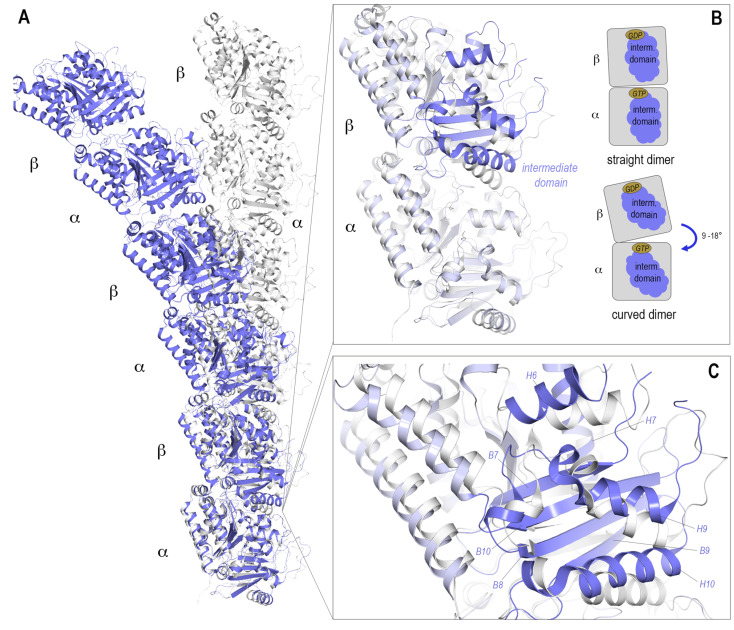
The “curved” and “straight” tubulin conformations. (**A**) A straight protofilament, as present in the MT lattice, is shown in ribbon presentation in light gray (PDB ID 7SJ7). A protofilament constituted of tubulin in a curved conformation is shown in blue (from PDB ID 5LXT). (**B**) The intrinsic curvature and structural differences on a single dimer are shown: A heterodimer in the straight conformation is depicted in light gray and the curved conformation in light blue. The main differences in the structures are within the intermediate domain (residues 206–384), highlighted in darker blue, which upon curved-to-straight transition moves relative to the other domains. This is also indicated in the schematic drawing of both straight and curved dimers. The angle corresponds to the relative curvature of one monomer to the other. (**C**) The structural elements of the intermediate domain are shown in more detail, the changes necessary for “straightening” are mainly translation of the shown H7 as well as rotation of the neighboring structural elements H6-10 and B7-10.

**Figure 3 biomolecules-13-00285-f003:**
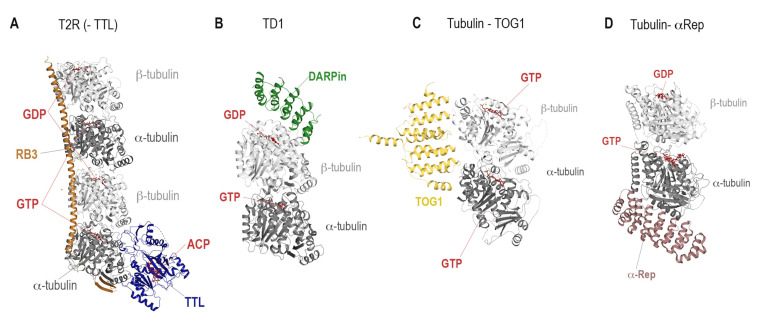
The crystallization systems (**A**) T_2_R(-TTL) (PDIB ID: 4I55, [44]), (**B**) TD1 (PDB ID 4DRX, [45], (**C**) Tubulin-TOG1 (PDB ID: 4FFB, [47]) and (**D**) Tubulin-αRep (PDB ID: 6GWC, [37]) are depicted. The proteins are shown in ribbon representation, α- and β-tubulin are colored dark and light grey, respectively. The SLD/RB3 protein is colored orange, the TTL in blue, DARPin in green, TOG1 in yellow and alpha-Rep in brownish color. Nucleotides are shown in sticks representation and colored red. The structure of the SLD tubulin complex, T_2_R crystallization system corresponds to the T_2_R-TTL structure without the bound TTL and thus was not shown separately.

**Figure 4 biomolecules-13-00285-f004:**
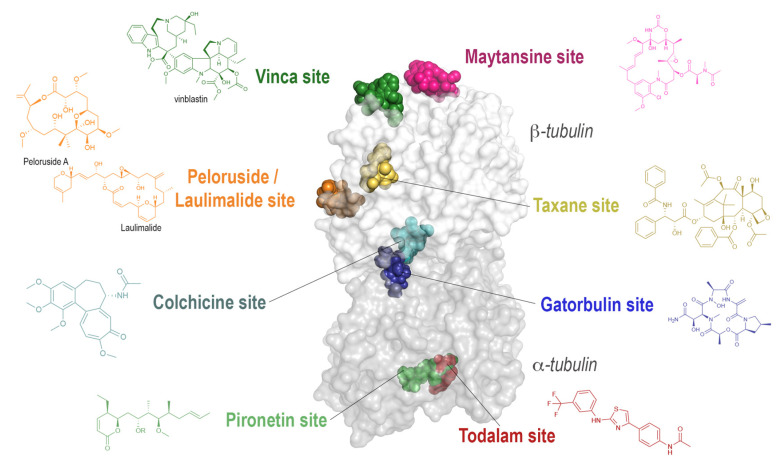
The eight distinct binding sites are highlighted on one tubulin dimer with all their representative ligands in colored sphere representation. The protein is shown in a transparent surface representation α- and β-tubulin chains are colored dark and light grey, respectively. The chemical structures of the ligands after which the binding sites were named are indicated next to the labels and colored following the color code of their sphere model.

**Table 1 biomolecules-13-00285-t001:** List of high-resolution tubulin crystal structures by binding site.

Binding Site	PDB ID	Resolution (Å)	Crystallization System	Bound Ligand
Apo	5NQU [68]	1.8	TD1	-
	3RYC [69]	2.1	T_2_R	-
	4I55 [44]	2.2	T_2_R-TTL	-
Taxane site	4I4T [44]	1.8	T_2_R-TTL	Zampanolide
	5LXT [70]	1.9	T_2_R-TTL	Discodermolide
	6SES [71]	2.0	T_2_R-TTL	B2
Laulimalide/Peloruside	4O4H [56]	2.1	T_2_R-TTL	Laulimalide
	4O4J [56]	2.2	T_2_R-TTL	Peroluside A
Maytansine	4TV9 [61]	2.0	T_2_R-TTL	PM060184
	6FJM [72]	2.1	T_2_R-TTL	Disorazole Z
	4TV8 [61]	2.1	T_2_R-TTL	Maytansine
Colchicine	6S8K [46]	1.5	TD1	Plinabulin
	6ZWB [73]	1.7	TD1	Z-SBTub3 photoswitch
	7Z2P [74]	2.0	T_2_R-TTL	Nocodazole
	5M7E [75]	2.0	T_2_R-TTL	BKM120
	6TH4 [76]	2.1	T_2_R	*exo*-methylene-nor-colchicine
Vinca	5IYZ [77]	1.8	T_2_R-TTL	Monomethylauristatin E
	5J2T [77]	2.2	T_2_R-TTL	Vinblastine
	5JH7 [67]	2.3	T_2_R-TTL	Eribulin
Pironetin	5LA6 [62]	2.1	T_2_R-TTL	Pironetin
	5FNV [63]	2.6	T_2_R-TTL	Pironetin
Todalam	5SB3 [12]	2.2	T_2_R-TTL	Todalam precursor 4
	5SB6 [12]	2.3	T_2_R-TTL	Todalam derivative 10
Gatorbulin	7ALR [64]	1.9	TD1	Gatorbulin

**Table 2 biomolecules-13-00285-t002:** High-resolution CryoEM MT structures.

MT Structure	PDBID	Resolution (Å)
Taxol-stablized MTs	6WVR [78]	2.9
Peloruside stabilized MTs	5SYC [55]	3.5
Taxol/Peloruside MTs	5SYE [55]	3.5
Taxol MTs	5SYF [55]	3.5
Zampanolide MTs	5SYG [55]	3.5
Undecorated MTs recombinant tubulin	7SJ7 [79]	3.8

**Table 3 biomolecules-13-00285-t003:** A glossary of key parameters and procedures used to analyze observed conformational changes during MD trajectories.

MD Analysis Metrics	Definition	Examples of Application
**RMSD**	The root mean square deviation (RMSD) is a standard measure of the structural distance between coordinates: it measures the average distance between a group of atoms. RMSD values help to evaluate the global structural stability of the system studied in the simulation.	Dash 2022 [119], El-Mernissi 2022 [112], Zhang 2022 [115], Zhao 2022 [120], Radha 2022 [123]
**RMSF**	The root mean square fluctuation (RMSF) represents the quadratic deviation of the atoms in temporal averages. RMSF values help to evaluate the internal structural flexibility of the studied system in the simulation.	Dash 2022 [119], El-Mernissi 2022, Zhang 2022 [115] Radha 2022 [123], Talimarada 2022 [124]
**Rg**	The radius of gyration (Rg) is defined as the mass-weighted root mean square atomic distance from the center-of-mass and can be applied to measure the level of structural compactness of a protein at different time points during the trajectory.	Hadizadeh 2022 [111], El-Mernissi 2022 [112], Zhang 2022 [115], Radha 2022 [123]. Rai 2022 [107]
**SASA**	The solvent accessible surface area (SASA) permits assessment of the overall changes in the tertiary structure of a molecule and its solvent accessibility over the course of the simulation.	El-Mernissi 2022 [112] Rai 2022 [107]
**2D interaction analysis**	2D interactions established between the protein and the ligand along the course of the simulations help to identify the residues within the binding site that play an important role in the binding of the ligand to the receptor and to list the ‘hot spots’ between the ligand and the protein.	Basu 2022 [121], Mao 2022 [90], Zhao 2022 [120], Rai 2022 [107], Zhang 2022 [103], Majumdar 2022 [125], Mao 2022 [90], Hadizadeh 2022 [111], Zhang 2022 [115]
**DSSP**	The Define Secondary Structure of Proteins (DSSP) algorithm is the standard method for assigning a secondary structure to amino acids of a protein given the atomic resolution coordinates of the protein.	Mao 2022 [90], Basu 2022 [121]
**Clustering**	Clustering is a data mining technique that allows molecular configurations to be grouped into subsets based on the similarity of their conformations.	Zhang 2022 [115]
**Binding free energy**	The Gibbs free energy (G) provides valuable information about the structure and stability of biomolecules. It is possible to calculate the predicted binding energy (ΔG_bind_) of a given tubulin-ligand complex using the MD simulation trajectory of this biomolecular association.	Zhao 2022 [120], Zhang 2022 [115], Elhemely 2022 [116], Rai 2022 [107], Radha 2022 [123], Majumdar 2019 [125]
**PRED**	The Per Residue Energy Decomposition (PRED) is a computational tool that is used to obtain the residue-wise contribution to the total binding free energy. It provides information on the key residues that contribute to protein-ligand association, the so-called ‘hot spots’.	Dash 2022 [119], Mao 2022 [90], Zhao 2022 [120], Zhang 2022 [120]
**CAS**	Computational Alanine Scanning (CAS) is a technique that consists of the mutation of amino acids present on the interaction surface between the protein and the ligand to alanine, and the measurement of the difference in binding free energy between the ligand and the native protein and the ligand and the multiple mutated proteins to identify ‘hot spots’.	Neto 2022 [118]
**PCA**	Principal Component Analysis (PCA) is a linear dimensionality reduction tool used in the MD field to map the coordinates of each frame of the trajectory to a linear combination of orthogonal vectors and to investigate the internal modes of motion of the system under study.	Basu 2022 [121]

## Data Availability

This is a review article, no new data was created.

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
