# Peer review of "Computational Approaches to the Rational Design of Tubulin-Targeting Agents"

_biomolecules, 2023, doi:10.3390/biom13020285_

Round 1
Reviewer 1 Report
The manuscript "Computational Approaches to the Rational Design of Tubulin Targeting Agents" is a thorough review about the most recent computational methods used for the design of new microtubule targeting agents. After exploring the structure of MTs, authors deeply describe both ligand and structure-based approaches, focusing on the different available binding site. The manuscript is well written and organized, there are no weaknesses and it presents useful data for the field. Hence it is suitable for publication in the present form.
Author Response
We wish to thank all reviewers for the positive comments and constructive criticism.
Best Regards,
Dragos Horvath
Reviewer 2 Report
The author summarizes the ligand and structure-based computational methods to study tubulin targeting agents with the available experimental structures of microtubules and tubulin. The manuscript is well-written and covers all technical aspects of designing in silico experiments. In addition, the author may want to consider adding limitations to an enhanced sample method for calculating free energy. A brief note on the accuracy of computationally predicted binding affinity of tubulin-targeting agents for tubulin compared with experimental evidence will be helpful.
Author Response
We wish to thank all reviewers for the positive comments and constructive criticism.
Q: In addition, the author may want to consider adding limitations to an enhanced sample method for calculating free energy.
A: Thank you for this comment. We agree that emphasizing the limitations of enhanced sampling techniques for calculating free energy adds great value to the text. In light of this, we have incorporated three brief paragraphs in the text dedicated to the limitations of each described enhanced sampling method. These paragraphs emphasize the importance of carefully choosing the appropriate collective variable when performing metadynamics [line 849-854] or reaction coordinates when applying umbrella sampling methods [lines 760-796], in order to accurately represent the physics of the system and avoid biases in the simulation results. Furthermore, we wish to emphasize that in the case of steered molecular dynamics, the force applied is not necessarily proportional to the binding free energy and this should be taken into account when interpreting the results obtained from this method. [lines 809-812]
Q: A brief note on the accuracy of computationally predicted binding affinity of tubulin-targeting agents for tubulin compared with experimental evidence will be helpful.
A: We have added a paragraph in the text noting that computational predictions of binding affinity can be very useful for rapidly and efficiently identifying potential binders and prioritizing them for further experimental validation but not to directly quantitatively correlate the binding affinity predicted computationally and experimental results. The success rate can vary depending on several factors, such as the quality of the computational method, the quality of the input data, and the complexity of the system being studied. [lines 608-620]
Best Regards,
Dragos Horvath
Reviewer 3 Report
In this work, Perez-Pena and colleagues summarize the computational approaches used to design microtubule targeting agents. The manuscript details latest advances in in silico drug design focusing on MTAs. This is a well written, timely and up to date review and it will be of interest for the readership of Biomolecules. Below are listed the comments that may be used for improving the manuscript prior to publication.
• In figure 1, the arrow indicating polymerization of tubulin heterodimers into MTs is in opposite direction.
• Figure 1 seems to be an exact recreation of a figure from Steinmetz et al., 2018. It would be nicer to create a more original figure in its place.
• While the authors describe 8 small molecule binding sites, it is reported as 7 in line 94.
• The references 35 and 36 are from 2014 and 2015, respectively. In order to incorporate the latest advancements in the context of the structure-based drug design, it may be useful to refer to more recent publications.
• In figure 2, the precise structures used should be cited along with their PDB in the figure legends. The “curved” and “straight” tubulin conformations shown in the figure do not highlight the curvature sufficiently. In addition to highlighting the conformational changes, it may be beneficial to the readers to use better visualization to highlight the curvature.
• While the authors mention five different systems for tubulin crystallization (line 330), they describe only four (tubulin-αRep is even termed as fourth in line 375). Hence, it would be more precise to be changed to four systems. Additionally, the authors fail to mention the tubulin/Zinc-sheets system. Although not widely employed, since it is the system that gave the first tubulin heterodimer structure, it deserves being mentioned.
• Typo in line 345: it should read “evolved to be the most commonly used”.
• In Figure 3, the PDB and references should be included for all the structures in the legend. Also, in the legends, its written that nucleotides are shown in red but since TOG1 is also shown in red, this may be misleading.
• Although the Tubulin-TOG1 and Tubulin-αRep systems are used to study microtubule binding proteins, to our knowledge, there are no published studies that have used these systems to study MTA binding. This may be relevant to mention, putting more focus on T2R(-TTL), TD1 systems.
• While the “Tools to Study Tubulin 3D Structures” section nicely describes the available crystallization systems, it might be beneficial to the readers if the authors could highlight the advantages and disadvantages of those systems for computational approaches. The readers may also benefit if the authors provide brief guidelines about when to use which system based on the MTAs being studied.
• Line 363: reference missing for the PDB 6S8K.
• Line 367: Ayaz et al. reference missing.
• In the “System selection for Virtual Screening (VS) and MD simulations” section, the authors claim that only a few structures have been used for simulations, the data/reference supporting this claim are missing.
• Lines 463, 466-468, 471 and 472: references missing for the PDB entries.
• In table 1, the PDB ID for Pironetin is incorrect. 4TUY is a tubulin structure with Rhizoxin.
• The authors should cite more recent reviews on sampling methods in addition to/instead of Meng et al. 578 [68] an Halperin et al. [69].
• The readers may benefit from an additional section on the use of machine learning in MTA studies.
Author Response
We wish to thank all reviewers for the positive comments and constructive criticism.
Q: In figure 1, the arrow indicating polymerization of tubulin heterodimers into MTs is in opposite direction.
A: Thank you for highlighting this erratum. In the new Figure 1 this has been fixed. [Figure 1 and figure caption, lines 82-88]
Q: Figure 1 seems to be an exact recreation of a figure from Steinmetz et al., 2018. It would be nicer to create a more original figure in its place.
A: We have replaced Figure 1 with a new figure, which presents a revised microtubule dynamics scheme that differs from the one presented in Steinmetz et al., 2018. Despite this change, we have ensured that all important information intended for the reader is still delivered through this new figure. [Figure 1 and figure caption, lines 82-88]
Q: The authors should cite more recent reviews on sampling methods in addition to/instead of Meng et al. 578 [68] an Halperin et al. [69].
A: We believe that Halperin et al. [69] is an essential reference to be left in as it gives a general overview over different docking approaches, algorithms, and underlying concepts. However, we agree that the readers may benefit from a more up-to-date overview of docking software and current developments in the field, as well as its place in modern structure-based virtual screening, hence we replaced a reference to Meng et al. [68] to a more recent review paper by Sulimov et al.
Q: While the authors describe 8 small molecule binding sites, it is reported as 7 in line 94.
A: In line 94 we referred to the 7 binding sites that were described before 2021, so before the publication of the crystallographic fragment screen (Mühlethaler et al 2021) or the rationally designed todalam, which represents the 8th distinct binding site on tubulin. We have added a sentence to clarify this point. [lines 100-101]
Q: In figure 2, the precise structures used should be cited along with their PDB in the figure legends. The “curved” and “straight” tubulin conformations shown in the figure do not highlight the curvature sufficiently. In addition to highlighting the conformational changes, it may be beneficial to the readers to use better visualization to highlight the curvature.
A: Indeed, we agree that the curvature could be better visualized; however it remains a challenging task. So far, the figure was mainly focused on displaying the structural differences of curved / straight tubulin within the dimer, as we thought this is of great concern for the selection of the structures for VS. We have revised the figure and included more visualization of the curvature on a more global scale to emphasize the overall effect and at the same time included some schemes to simplify the concept, while retaining the important aspects of the differences on a single dimer one needs to be aware of when choosing the conformation for VS. [exchanged figure 2, lines 358-366]
Q: While the authors mention five different systems for tubulin crystallization (line 330), they describe only four (tubulin-αRep is even termed as fourth in line 375). Hence, it would be more precise to be changed to four systems. Additionally, the authors fail to mention the tubulin/Zinc-sheets system. Although not widely employed, since it is the system that gave the first tubulin heterodimer structure, it deserves being mentioned.
A: Thank you for pointing out this discrepancy, we have adjusted the text to clarify that we consider T2R, T2R-TTL, TD1, tubulin-TOG1 and tubulin alphaRep as 5 different crystal systems. Although the T2R-TTL system clearly evolved from the T2R method, the packing of the molecules in the crystals differ substantially, so we consider them both as well independently. We agree that the tubulin zinc sheets system deserved to be recognized as being the first method to give any structural information on tubulin and have added a small paragraph to highlight this achievement. [lines 419-426]
Q: Typo in line 345: it should read “evolved to be the most commonly used”.
Fixed, thank you.
Q: In Figure 3, the PDB and references should be included for all the structures in the legend. Also, in the legends, its written that nucleotides are shown in red but since TOG1 is also shown in red, this may be misleading.
A: We have provided now all the individual PDB codes and references for the structures shown, as well as changed the color for the TOG1 as was suggested. [figure 3 and figure caption]
Q: Although the Tubulin-TOG1 and Tubulin-αRep systems are used to study microtubule binding proteins, to our knowledge, there are no published studies that have used these systems to study MTA binding. This may be relevant to mention, putting more focus on T2R(-TTL), TD1 systems.
To our knowledge the two mentioned systems have not been used to study small molecule tubulin interactions, we have included them in case they might be more relevant to the field at a later time and to provide a more comprehensive overview of all recently used crystallization systems. We have added a sentence to emphasize this to the reader and to highlight the T2R(-TTL) and TD1 systems. [lines 462-466]
Q: While the “Tools to Study Tubulin 3D Structures” section nicely describes the available crystallization systems, it might be beneficial to the readers if the authors could highlight the advantages and disadvantages of those systems for computational approaches. The readers may also benefit if the authors provide brief guidelines about when to use which system based on the MTAs being studied.
A: We have added a comment on which system should be considered for reach binding site – however we kept it brief since a broader explanation would be out of the scope of the Review which aims to summarize the basic structural data available as a basis for computational analysis. Further, we have added some more specific guidelines and recommendations for individual sites to the system selection for VS part and cautioned the readers to evaluate each targeted site on the impact of the stabilizing protein or the crystal packing. [lines 620-648, 728-733, 743-747]
Q: Line 363: reference missing for the PDB 6S8K.
A: We have included the reference for the PDB entry.
Q: Line 367: Ayaz et al. reference missing.
We have added the missing reference to the list.
Q: In the “System selection for Virtual Screening (VS) and MD simulations” section, the authors claim that only a few structures have been used for simulations, the data/reference supporting this claim are missing.
A: We have noticed these trends during our preparations for this Review, specifically while analyzing the structures that we used for MD simulations. We have added a link to the table A6 in which the data is summarized and rephrased the paragraph. [lines 652-654, 668-675]
Q: Lines 463, 466-468, 471 and 472: references missing for the PDB entries.
A: We have added the corresponding references to the PDB entries. [lines 658-667]
Q: In table 1, the PDB ID for Pironetin is incorrect. 4TUY is a tubulin structure with Rhizoxin.
A: Thank you for highlighting this erratum. We have corrected it to the right PDB entry and as well added all the references. [Table 1]
Q: The references 35 and 36 are from 2014 and 2015, respectively. In order to incorporate the latest advancements in the context of the structure-based drug design, it may be useful to refer to more recent publications.
A: Thank you for your comment. We accounted for it by introducing references to more recent review articles on structure-based drug design instead of references 35 and 36, now references 88 and 89 are in place.
Q: The readers may benefit from an additional section on the use of machine learning in MTA studies.
A : A paragraph was added to highlight an important issue in tubulin-related ligand-based design, which is the fact that the absence of large and diverse structure-based data on tubulin polymerization modulators make it impossible to utilize the full potential of machine learning yet. Based on a ChEMBL query, the (disappointing) status quo of publicly available structure-tubulin activity “machine-learnable” sets is briefly discussed. [lines 221-242]
Best Regards,
Dragos Horvath